# Prioritizing Cleaner Production Actions towards Circularity: Combining LCA and Emergy in the PET Production Chain

**Gustavo Bustamante** [1], **Biagio Fernando Giannetti** [1,2,*], **Feni Agostinho** [1,2], **Gengyuan Liu** [2,3] and **Cecília M. V. B. Almeida** [1,2]

1   Laboratório de Produção e Meio Ambiente, Programa de Pós-Graduação em Engenharia de Produção, Universidade Paulista, R. Dr. Bacelar 1212, São Paulo 04026-002, Brazil; gustamante1000@hotmail.com (G.B.); feni@unip.br (F.A.); cmvbag@unip.br (C.M.V.B.A.)

2   State Key Joint Laboratory of Environment Simulation and Pollution Control, School of Environment, Beijing Normal University, Beijing 100875, China; liugengyuan@bnu.edu.cn

3   Beijing Engineering Research Center for Watershed Environmental Restoration & Integrated Ecological Regulation, Beijing 100875, China

*   Correspondence: biafgian@unip.br

**Abstract:** Petrochemicals, which convert oil and gas into products such as plastics, are fundamental to modern societies. Chemists recognize their role in designing materials and the adverse effects that these may have on the environment, preventing sustainable development. Several methodological frameworks and sustainability assessment approaches have been developed to evaluate the resources used in the petrochemical sector in terms of environmental costs. Still, there is a need to evaluate these systems in terms of environmental costs deeply. A combination of life cycle assessment and emergy accounting—to assess the environmental support for resource use—is applied in this study of the PET production chain in Europe. The unit emergy values of several intermediates are calculated or updated to facilitate the discernment of the quality of energy used and the processes' efficiency. Several routes for synthesizing renewable para-xylene and ethylene glycol from biomass are discussed and confronted with the efforts focused on recycling and recovering the final product, providing concurrently a procedure and a valuable data set for future CP actions. The results show that understanding the efficiencies changing across the production chain may help stakeholders decide where and when interventions to promote a circular economy are most effective along a petrochemical production chain.

**Keywords:** resource efficiency; productive chain trade-offs; decision making

## 1. Introduction

Modern industry requires different inputs to support production and profit. In particular, the chemistry sector faces numerous challenges as it contributes to almost all the raw materials transformations into the finished products. The sector's decisions directly affect the exploitation of natural resources and generate pollution with consequences to humanity at the societal/environmental dimensions [1]. In parallel, the current economy is still based on a linear model in which resources are extracted according to the production demand and consumption patterns. In addition, plastics production is more and more challenged to cater to an increasing population with nonrenewable resources while controlling the adverse impacts on the environmental capacity of support. Such an economic system is indubitably not sustainable. Global consensus has already identified it as a major cause of local and global problems such as food/energy/water scarcity, climate change, and biodiversity loss [2,3]. It is urgent to let sustainability awareness and economy evolve together [4,5] to ensure that finite resource exploitation occurs under the best conditions allowed by the current technological/methodological development level.

A circular economy (CE) can play several crucial roles under this standpoint as a stimulating economic model that may help to preserve materials and products with high utility and value [4]. Geissdoerfer et al. [6] highlighted the CE to be a required posture toward sustainability, as it may impose a regenerative structure that minimizes material and energy use, emissions, and wastes, by decelerating and closing the resource loops, the two essential practices that make the CE different from the linear economy [7]. In particular, slowing happens when long-life goods and product-life extension solutions are designed, and the use time of products is extended and/or intensified, contributing to slowing down the resource flows. When loops between end-of-life and production are closed, postuse products are recirculated within the life cycle, and low-burden resources are used to produce secondary raw materials [7]. Through those two practices, a CE may satisfy the principle that the planet is a closed system with limited ability to support the increasing population growth and tolerate environmental degradation [8,9]. The economy and the environment need, therefore, to coexist to preserve the earth's carrying capacity [8,9] by adopting, implementing, and spreading sustainability-oriented strategies [10] and to effectively help the transition to a carbon-neutral CE [10,11].

The petrochemical sector is responsible for diverse environmental impacts that depend upon the amount of nonrenewable resources usage and the volume of pollutants released, damaging if directly discharged into the environment, even without an immediate impact [12]. These pollutants can, for example, adversely affect the soil, water, and microbial populations present in the soil system. Later on, the nutrient cycling processes would be affected because microbial populations are responsible for disintegrating organic matter [13–15] and enhancing the nutrient cycling [16–18]. Due to the reduced nutrient cycling, crop production would be decreased [19].

Along the petrochemical production chains, nature and the human systems invest materials, energy, and services in each petrochemical product supplied for societal use. The plastics chain was recognized in [10] as posing sustainability challenges that demand serious, all-inclusive, and organized CE-based actions. The EU Strategy for Plastics has developed a comprehensive set of initiatives to respond to the pressing global need to deal with plastics' end-of-life and to governments and companies pursuing sustainability goals using CE. Under this perspective, evaluating and improving the relevant environmental issues associated with plastics' life cycle may help meet such purposes [10].

The most common polyester among plastics is polyethylene terephthalate (PET), with nearly 70 million tons supplied yearly for packaging and textiles [20]. Several contributions to altering the linear resource-depleting industry to a resource-valuing one have risen [21], especially regarding recycled PET to replace petrochemical PET, contributing toward a circular PET economy. There are options such as waste-to-material through mechanical recycling [22,23], chemical recycling [24–26], and decomposition into fibers or smaller molecular components using sub- and supercritical fluids [27,28] or hydrolyzing enzymes [29]. There are also proposals to use flakes made of postuse bottle-derived PET as an insulating material for building thermal panels [30] and other construction materials [31]. Waste plastic has developed into one of the most concerning environmental problems [32] and a central issue in waste management [33], considering the growth in production and fast consumption of the final products [34]. Thus, most efforts to insert the PET chain into the circular economy are focused on recycling and recovering the final product. However, these efforts to improve circularity are still marked by complexities and uncertainties [35], which should be managed and accounted for by practical and multidisciplinary tools that allow proper certification [36] and creating standards and labels to ensure accurate decision-making. There is a need for assessing quantitatively if the solutions proposed to make the petrochemical sector chain circular effectively reduce the environmental burden or if they only alleviate the disposal-side and cause damages in other parts of the production chain or other parts of the environment. The investment to improve circularity, reducing the use of energy and materials, may be discriminated against and accounted for by applying assessment methods capable of supplying quantitative indications to decision-makers.

Sustainability evaluation methods may help decision-makers ponder upon informed cleaner production (CP) choices on ecologically responsible procedures in production chains, mainly when their activities occur at the society/nature interface. The sustainability assessments supported by systematical and rigorous scientific frameworks and wide-ranging methods such as LCA (life cycle assessment) and EmA (emergy accounting) are necessary to assist with environmental and economic sustainability.

LCA ponders on the relation between processes and the environment, which acts as a sink and a source and is based on inventories to develop indices connected to impact categories such as global warming. This method has been widely used to estimate the environmental load inflicted by by-products and processes of the petrochemical industry [37,38], and the most complete and organized source of data on this sector is available on the Plastics Europe website, a pan-European association that represents European manufacturers. The LCA-based ecoprofiles [39,40] provide information on market data, education, and publications, but the authors also focused on the development of mathematical models to optimize petrochemical complexes by combining economic and environmental criteria [41,42] and the effects of replacing PET as packaging material [43,44].

EmA associates resource inflows to the unit emergy values (UEVs) that evidence the requirement for nature's support per resource unit. UEVs can be used to value and compare products on the same basis [45] and are useful indicators to estimate the environmental efficiency of production processes. Accounting for the work performed by the biosphere to produce a resource, the UEVs inform the extent to which the environmental support for natural resource generation converts into the final product. Lower UEVs suggest a higher resource use efficiency [46]. EmA also appeared as an option to assess the environmental performance of the petrochemical sector, accounting for the contributions from nature and from the man-made production chain evidencing the quantities of energy and materials, directly or indirectly, required to obtain products and services [45]. Oil-based production routines were evaluated by Bastianoni et al. [47], who calculated the unit emergy values (UEVs) of natural gas and crude oil, and subsequently, the UEVs of the liquid petroleum products [48], which were used to estimate the emergy contribution of naphtha for different ethylene production processes [49]. Brown et al. [50] divided crude oil and natural gas production into two formation phases: organic matter generated by tidal, solar, and geothermal energies and petroleum generated by geothermal energy. The calculated UEVs give an idea of the environmental load imposed by exploiting these resources on the donor-side, that is, the available energy required to supply these resources to the chemical industry and, ultimately, to societal use.

Previous efforts to integrate EmA and LCA combined the extensive LCA inventories and rules with the more far-reaching conceptual framework of the EmA approach to make environmentally sensible decisions in engineering processes that would benefit from combining the two methods [51]. However, the effort to integrate structure and software showed some barriers challenging to overcome, such as conflicting specific goals, allocation rules, and spatial and time scales [52–55], although authors agreed that EmA could benefit by expanding the accessible LCA databases [56] and permitting a broader understanding of the existing systems and their potential improvements scenarios [57].

Under the circular viewpoint, life cycle thinking and emergy/exergy evaluations were considered suitable to subsidize the chemical sector for developing innovative value-added chemical products produced from waste-derived materials [58]. In this context, Almeida et al. [59] applied the emergy synthesis combined with LCI data on the aluminum and PET packaging life cycles, including the recycling options contributing to the processes and materials selection during the product design stage or to establish public policies for recycling options [60]. UEVs close to those calculated by Bastianoni et al. [48] and Brown et al. [50] were obtained by an LCI-based unit emergy values of naphtha calculated by Bustamante et al. [61].

The literature review suggests that combining EmA and LCA may improve the evaluations of the environmental performance and the choice of CP actions by (1) integrating

opposed perspectives: the donor/supplier-side with a user/receiver-side and (2) including EmA's concept of the environmental work necessary to obtain a product escalating the spatial and time scales of LCA [51].

In this work, a course of action combining EmA and LCA provides an additional step toward assessing and documenting procedures for the CE, considering some of the complexities and uncertainties involved in sustainability assessments. Results thus contribute by (1) underscoring the use of LCA inventories to contribute to the CE; (2) providing new values of the UEVs of PET and intermediate chemicals and facilitating the discernment of the quality of energy used and the processes' efficiency; (3) providing an alternate method for calculating UEVs using an LCA and EmA combination through the investigation of the PET resin production chain; (4) helping to prioritize actions towards closing loops to achieve a cleaner/circular petrochemical industry.

## 2. Methods

LCA is a well-recognized assessment method for registering and assessing inputs, outputs, and the prospective impacts over its entire life cycle [62,63]. The LCA structure includes the definition of scope, inventory, impact assessment, and interpretation. For the interpretation phase, the method offers indicators associated with categories of environmental impacts, such as climate change, acidification, and land use.

Raw data of the PET production chain were collected using the LCI databases from Plastics Europe [40,64,65], which include all flows that contribute to the processes from the extraction of natural resources to obtaining 1 kg of product "at gate" set to be used in the next stage (representing the average of European industrial production) or to obtain 1 kg of finished product ready for delivery to consumers/users. Therefore, the LCI database and the ecoprofiles follow the instructions of the ISO 14040-44 standards series [40,64,65].

Following the growing attention on the embodied energy accounting system and the interface energy/environment [66–68], EmA was introduced by H.T. Odum to express the work done by nature to obtain every biosphere element, with the idea of expressing all resource flows in the same unit (solar energy joules, seJ). Emergy is the available energy directly or indirectly required to generate a product or service [45]. The total emergy is calculated by the sum of each input's energy content multiplied by its UEV (seJ/unit), which denotes the emergy invested to generate one flow unit. UEVs refer to a geobiosphere emergy baseline ($12.0 \times 10^{24}$ seJ/year) that results from the sum of the annual emergies—solar, gravitational, and geothermal—that drive the biosphere [69] and were calculated by dividing the total emergy required to produce a given output by one unit of this output.

*Calculating UEVs from LCIs of the Explored Petrochemicals*

Regarding the target products, it was considered that PET is most commonly produced by the polycondensation reaction between purified terephthalic acid obtained from para-xylene—made from monomers of benzene, toluene, xylene isomers (BTX), and ethylene glycol [70]—produced from ethylene by the steam cracking of hydrocarbons through the ethylene oxide intermediate [71].

The investigated chemicals were tracked back using the Plastics Europe [40,65] inventories to recover the original inputs of the PET production chain. These ecoprofiles follow the instructions for calculating the information according to the ISO 14040-44 standards [40,65].

The premise for the UEVs calculation for each output was established considering no allocation [72]—accounting for all flows of natural resources contributing to obtaining 1 kg of product "at gate", that is, inventories were directly used to convert LCIs raw data into emergy values. In this way, the emergy calculations agreed with the standards of the LCA procedure (Equations (1) and (2)).

$$U = \sum E_i \times UEV_i \qquad (1)$$

where U is the total emergy (seJ), $E_i$ is the available energy of the ith input flow—material and energy delivered to the system. $UEV_i$ is the emergy required to obtain one unit of that

input flow (Equation (2)). The quantity of every input was multiplied by its corresponding UEV resulting in the emergy needed to produce 1 kg of output.

$$UEV_j = U/Y_j \tag{2}$$

where $UEV_j$ is the UEV of the product j, U is the total emergy, and $Y_j$ is the yield of the process, that is, the product's energy or mass content.

The production flow involving the nine investigated chemicals (crude oil, naphtha, xylene, ethylene, p-xylene, ethylene oxide, ethylene glycol, purified terephthalic acid, and PET) is shown in Figure 1 in a conventional flow chart. The complete tables for all materials are available in Supplementary Materials, Tables S1–S8.

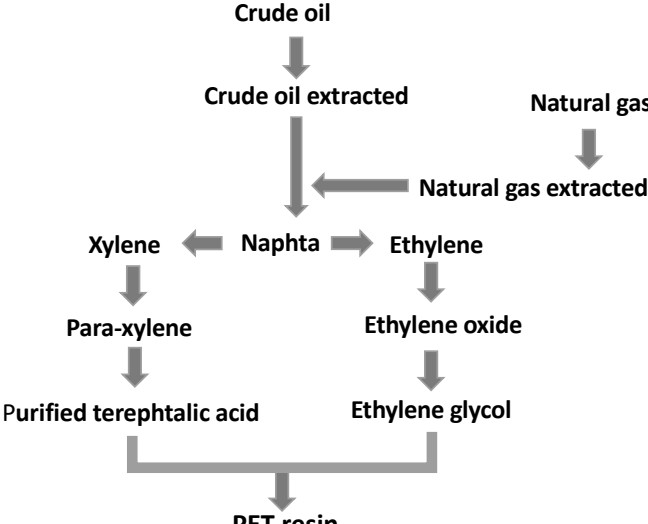

**Figure 1.** The production flow of PET resin involving the nine investigated intermediate inputs (crude oil, naphtha, xylene, ethylene, p-xylene, ethylene oxide, ethylene glycol, and purified terephthalic acid).

The LCIs include contributions from different material flows, energy, fuels, and metals for each chemical product, among others. The resulting emergy tables estimate the energy directly or indirectly used to obtain these flows. The flows relative to the facilities' construction stage and maintenance were not accounted for because of their low influence on emergy results [48,49]. Similarly, entries for labor and services were not included because these flows are country-specific and depend upon local economic and social structures. When labor and services are included, results also become country-specific, and the overall scope of evaluating the technical processes becomes compromised [73].

## 3. Results

Results are presented in two steps: (Section 3.1) the calculation of LCI-based UEVs and (Section 3.2) the potential use of UEVs to support decision-making toward a circular economy.

### 3.1. Calculations and Analysis of the LCI-Based UEVs

The LCI-based UEVs of the PET production and nine selected intermediates (crude oil, naphtha, xylene, ethylene, p-xylene, ethylene oxide, ethylene glycol, and purified terephthalic acid—with detailed calculations in the Supplementary Material, Tables S1–S8) are shown and compared with UEVs retrieved from previously published papers.

The LCI-based emergy for PET is shown in Table 1. Local renewable resource flows provide less than 1% of the total emergy of the PET production. The resulting UEV of PET is $5.94 \times 10^9$ seJ/g, with crude oil contributing the most (71.4%), followed by natural gas (19.2%).

**Table 1.** LCI-based * emergy required to produce 1 kg of PET resin.

| | INPUTS | Unit | Quantity * | UEV ** | Unit | Emergy |
|---|---|---|---|---|---|---|
| 1 | Energy, gross calorific value, in biomass | MJ | $4.22 \times 10^{-1}$ | $6.75 \times 10^{4}$ | seJ/J | $2.85 \times 10^{10}$ |
| 2 | Peat, in ground | kg | $9.39 \times 10^{-7}$ | $3.19 \times 10^{4}$ | seJ/J | $2.92 \times 10^{5}$ |
| 3 | Wood, primary forest, standing | m3 | $1.04 \times 10^{-6}$ | $1.04 \times 10^{4}$ | seJ/J | $1.22 \times 10^{8}$ |
| 5 | Energy, gross calorific value, renewable | MJ | $4.55 \times 10^{-1}$ | $1.02 \times 10^{5}$ | seJ/J | $4.63 \times 10^{10}$ |
| 6 | Energy, hydro | MJ | $3.39 \times 10^{-1}$ | $1.35 \times 10^{5}$ | seJ/J | $4.56 \times 10^{10}$ |
| 7 | Energy, gross calorific value, in lignite | MJ | $1.39 \times 10^{0}$ | $6.22 \times 10^{4}$ | seJ/J | $8.63 \times 10^{10}$ |
| 8 | Aluminium, 24% in bauxite, 11% in crude ore, in ground | kg | $1.14 \times 10^{-5}$ | $5.40 \times 10^{9}$ | seJ/g | $6.15 \times 10^{7}$ |
| 9 | Barite, 15% in crude ore, in ground | kg | $2.67 \times 10^{-5}$ | $1.68 \times 10^{9}$ | seJ/g | $4.49 \times 10^{7}$ |
| 10 | Basalt, in ground | kg | $3.61 \times 10^{-10}$ | $7.56 \times 10^{9}$ | seJ/g | $2.73 \times 10^{3}$ |
| 11 | Borax, in ground | kg | $1.63 \times 10^{-10}$ | $1.68 \times 10^{9}$ | seJ/g | $2.75 \times 10^{2}$ |
| 12 | Cadmium, 0.30% in sulfide, Cd 0.18%, Pb, Zn, Ag, In, in ground | kg | $6.54 \times 10^{-12}$ | $3.40 \times 10^{13}$ | seJ/g | $2.22 \times 10^{5}$ |
| 13 | Calcite, in ground | kg | $9.02 \times 10^{-3}$ | $1.68 \times 10^{9}$ | seJ/g | $1.51 \times 10^{10}$ |
| 14 | Carbon, in organic matter, in soil | kg | $3.34 \times 10^{-8}$ | $2.77 \times 10^{9}$ | seJ/g | $9.25 \times 10^{4}$ |
| 15 | Chromium, 25.5% in chromite, 11.6% in crude ore, in ground | kg | $1.13 \times 10^{-5}$ | $1.50 \times 10^{11}$ | seJ/g | $1.70 \times 10^{9}$ |
| 16 | Chrysotile, in ground | kg | $2.97 \times 10^{-7}$ | $1.68 \times 10^{9}$ | seJ/g | $4.98 \times 10^{5}$ |
| 17 | Clay, unspecified, in ground | kg | $1.95 \times 10^{-4}$ | $4.80 \times 10^{9}$ | seJ/g | $9.39 \times 10^{8}$ |
| 18 | Coal | MJ | $2.45 \times 10^{0}$ | $5.71 \times 10^{4}$ | seJ/J | $1.40 \times 10^{11}$ |
| 19 | Cobalt, in ground | kg | $1.80 \times 10^{-4}$ | $1.30 \times 10^{11}$ | seJ/g | $2.34 \times 10^{10}$ |
| 20 | Colemanite, in ground | kg | $4.53 \times 10^{-9}$ | $1.68 \times 10^{9}$ | seJ/g | $7.62 \times 10^{3}$ |
| 21 | Copper, in ground | kg | $1.07 \times 10^{-6}$ | $9.80 \times 10^{10}$ | seJ/g | $1.05 \times 10^{8}$ |
| 22 | Diatomite, in ground | kg | $2.00 \times 10^{-13}$ | $1.68 \times 10^{9}$ | seJ/g | $3.36 \times 10^{-1}$ |
| 23 | Dolomite, in ground | kg | $2.53 \times 10^{-6}$ | $1.85 \times 10^{10}$ | seJ/g | $4.69 \times 10^{7}$ |
| 24 | Feldspar, in ground | kg | $2.07 \times 10^{-12}$ | $1.68 \times 10^{9}$ | seJ/g | $3.48 \times 10^{0}$ |
| 25 | Fluorine, in ground | kg | $8.13 \times 10^{-6}$ | $1.68 \times 10^{9}$ | seJ/g | $1.37 \times 10^{7}$ |
| 26 | Fluorspar, 92%, in ground | kg | $8.37 \times 10^{-6}$ | $8.38 \times 10^{8}$ | seJ/g | $7.02 \times 10^{6}$ |
| 27 | Gas, natural, in ground | MJ | $1.67 \times 10^{1}$ | $6.83 \times 10^{4}$ | seJ/J | $1.14 \times 10^{12}$ |
| 28 | Gold, in ground | kg | $2.44 \times 10^{-17}$ | $5.00 \times 10^{11}$ | seJ/g | $1.22 \times 10^{-2}$ |
| 29 | Granite, in ground | kg | $2.07 \times 10^{-12}$ | $8.40 \times 10^{8}$ | seJ/g | $1.74 \times 10^{0}$ |
| 30 | Gravel, in ground | kg | $6.65 \times 10^{-3}$ | $8.40 \times 10^{8}$ | seJ/g | $5.59 \times 10^{9}$ |
| 31 | Gypsum, in ground | kg | $3.19 \times 10^{-9}$ | $2.85 \times 10^{9}$ | seJ/g | $9.10 \times 10^{3}$ |
| 32 | Indium, 0.005% in sulfide, In 0.003%, Pb, Zn, Ag, Cd, in ground | kg | $1.03 \times 10^{-13}$ | $4.03 \times 10^{11}$ | seJ/g | $4.15 \times 10^{1}$ |
| 33 | Iron, 46% in ore, 25% in crude ore, in ground | kg | $4.18 \times 10^{-5}$ | $1.20 \times 10^{10}$ | seJ/g | $5.02 \times 10^{8}$ |
| 34 | Kaolinite, 24% in crude ore, in ground | kg | $6.87 \times 10^{-7}$ | $1.68 \times 10^{9}$ | seJ/g | $1.15 \times 10^{6}$ |
| 35 | Kieserite, 25% in crude ore, in ground | kg | $1.69 \times 10^{-9}$ | $1.68 \times 10^{9}$ | seJ/g | $2.84 \times 10^{3}$ |
| 36 | Lead, 5.0% in sulfide, Pb 3.0%, Zn, Ag, Cd, In, in ground | kg | $4.31 \times 10^{-11}$ | $4.80 \times 10^{11}$ | seJ/g | $2.07 \times 10^{4}$ |
| 37 | Magnesite, 60% in crude ore, in ground | kg | $6.86 \times 10^{-7}$ | $1.68 \times 10^{9}$ | seJ/g | $1.15 \times 10^{6}$ |
| 38 | Manganese, 35.7% in sedimentary deposit, 14.2% in crude ore, in ground | kg | $3.15 \times 10^{-4}$ | $3.50 \times 10^{11}$ | seJ/g | $1.10 \times 10^{11}$ |
| 39 | Mercury, in ground | kg | $2.71 \times 10^{-8}$ | $4.20 \times 10^{13}$ | seJ/g | $1.14 \times 10^{9}$ |
| 40 | Metamorphous rock, graphite containing, in ground | kg | $8.31 \times 10^{-9}$ | $1.68 \times 10^{9}$ | seJ/g | $1.39 \times 10^{4}$ |
| 41 | Molybdenum, 0.025% in sulfide, Mo $8.2 \times 10^{-3}$% and Cu 0.39% in crude ore, in ground | kg | $4.75 \times 10^{-7}$ | $7.00 \times 10^{11}$ | seJ/g | $3.32 \times 10^{8}$ |
| 42 | Nickel, 1.13% in sulfide, Ni 0.76% and Cu 0.76% in crude ore, in ground | kg | $2.63 \times 10^{-5}$ | $2.00 \times 10^{11}$ | seJ/g | $5.26 \times 10^{9}$ |
| 43 | Oil, crude, in ground | MJ | $4.49 \times 10^{1}$ | $9.45 \times 10^{4}$ | seJ/J | $4.24 \times 10^{12}$ |
| 44 | Olivine, in ground | kg | $7.05 \times 10^{-10}$ | $1.68 \times 10^{9}$ | seJ/g | $1.19 \times 10^{3}$ |
| 45 | Pd, in ground | kg | $1.13 \times 10^{-8}$ | $1.20 \times 10^{11}$ | seJ/g | $1.36 \times 10^{6}$ |
| 46 | Phosphorus, 18% in apatite, 12% in crude ore, in ground | kg | $3.25 \times 10^{-5}$ | $2.07 \times 10^{10}$ | seJ/g | $6.71 \times 10^{8}$ |
| 47 | Potassium chloride, in ground | kg | $4.02 \times 10^{-8}$ | $4.97 \times 10^{9}$ | seJ/g | $2.00 \times 10^{5}$ |
| 48 | Pt, in ground | kg | $1.95 \times 10^{-12}$ | $3.70 \times 10^{11}$ | seJ/g | $7.20 \times 10^{2}$ |
| 49 | Rh, in ground | kg | $1.74 \times 10^{-12}$ | $1.20 \times 10^{12}$ | seJ/g | $2.09 \times 10^{3}$ |
| 50 | Rhenium, in crude ore, in ground | kg | $4.83 \times 10^{-13}$ | $8.93 \times 10^{12}$ | seJ/g | $4.31 \times 10^{3}$ |
| 51 | Rutile, in ground | kg | $2.07 \times 10^{-12}$ | $1.68 \times 10^{9}$ | seJ/g | $3.48 \times 10^{0}$ |
| 52 | Sand, unspecified, in ground | kg | $5.61 \times 10^{-8}$ | $1.68 \times 10^{9}$ | seJ/g | $9.43 \times 10^{4}$ |
| 53 | Shale, in ground | kg | $6.13 \times 10^{-9}$ | $1.68 \times 10^{9}$ | seJ/g | $1.03 \times 10^{4}$ |
| 54 | Silver, in ground | kg | $6.51 \times 10^{-17}$ | $4.50 \times 10^{11}$ | seJ/g | $2.93 \times 10^{-2}$ |
| 55 | Sodium bromide, in ground | kg | $4.40 \times 10^{-4}$ | $1.68 \times 10^{9}$ | seJ/g | $7.39 \times 10^{8}$ |
| 56 | Sodium chloride, in ground | kg | $1.30 \times 10^{-2}$ | $1.68 \times 10^{9}$ | seJ/g | $2.19 \times 10^{10}$ |
| 57 | Sodium nitrate, in ground | kg | $3.33 \times 10^{-15}$ | $1.68 \times 10^{9}$ | seJ/g | $5.60 \times 10^{-3}$ |
| 58 | Sodium sulphate, various forms, in ground | kg | $2.24 \times 10^{-6}$ | $1.40 \times 10^{9}$ | seJ/g | $3.13 \times 10^{6}$ |
| 59 | Stibnite, in ground | kg | $2.71 \times 10^{-4}$ | $1.68 \times 10^{9}$ | seJ/g | $4.55 \times 10^{8}$ |

**Table 1.** *Cont.*

| | INPUTS | Unit | Quantity * | UEV ** | Unit | Emergy |
|---|---|---|---|---|---|---|
| 60 | Sulfur, in ground | kg | $4.60 \times 10^{-5}$ | $2.08 \times 10^{10}$ | seJ/g | $9.58 \times 10^{8}$ |
| 61 | Talc, in ground | kg | $9.79 \times 10^{-8}$ | $2.80 \times 10^{10}$ | seJ/g | $2.74 \times 10^{6}$ |
| 62 | Tantalum, 81.9% in tantalite, $1.6 \times 10^{-4}$% in crude ore, in ground | kg | $2.36 \times 10^{-17}$ | $1.70 \times 10^{11}$ | seJ/g | $4.02 \times 10^{-3}$ |
| 63 | Tellurium, 0.5 ppm in sulfide, Te 0.2 ppm, Cu and Ag, in crude ore, in ground | kg | $3.20 \times 10^{-18}$ | $5.04 \times 10^{13}$ | seJ/g | $1.61 \times 10^{-1}$ |
| 64 | Tin, 79% in cassiterite, 0.1% in crude ore, in ground | kg | $3.01 \times 10^{-11}$ | $1.70 \times 10^{12}$ | seJ/g | $5.11 \times 10^{4}$ |
| 65 | TiO2, 54% in ilmenite, 2.6% in crude ore, in ground | kg | $4.89 \times 10^{-6}$ | $3.82 \times 10^{10}$ | seJ/g | $1.87 \times 10^{8}$ |
| 66 | Ulexite, in ground | kg | $1.21 \times 10^{-17}$ | $1.68 \times 10^{9}$ | seJ/g | $2.02 \times 10^{-5}$ |
| 67 | Uranium, in ground | kg | $6.78 \times 10^{-6}$ | $1.60 \times 10^{11}$ | seJ/g | $1.08 \times 10^{9}$ |
| 68 | Zinc, 9.0% in sulfide, Zn 5.3%, Pb, Ag, Cd, In, in ground | kg | $8.11 \times 10^{-7}$ | $7.20 \times 10^{10}$ | seJ/g | $5.84 \times 10^{7}$ |
| 69 | Zirconium, 50% in zircon, 0.39% in crude ore, in ground | kg | $3.25 \times 10^{-17}$ | $3.18 \times 10^{10}$ | seJ/g | $1.03 \times 10^{-3}$ |
| 70 | Magnesium, 0.13% in water | kg | $3.73 \times 10^{-18}$ | $1.68 \times 10^{9}$ | seJ/g | $6.26 \times 10^{-6}$ |
| 71 | Water, cooling, unspecified natural origin | $m^3$ | $5.12 \times 10^{-2}$ | $2.70 \times 10^{5}$ | seJ/g | $1.39 \times 10^{10}$ |
| 72 | Water, lake | $m^3$ | $1.52 \times 10^{-3}$ | $4.52 \times 10^{5}$ | seJ/g | $6.85 \times 10^{8}$ |
| 73 | Water, process, unspecified natural origin | $m^3$ | $4.20 \times 10^{-3}$ | $6.74 \times 10^{4}$ | seJ/J | $1.18 \times 10^{9}$ |
| 74 | Water, river | $m^3$ | $1.77 \times 10^{-3}$ | $3.41 \times 10^{5}$ | seJ/g | $6.03 \times 10^{8}$ |
| 75 | Water, salt, ocean | $m^3$ | $5.38 \times 10^{-5}$ | $5.36 \times 10^{4}$ | seJ/J | $1.21 \times 10^{7}$ |
| 76 | Water, salt, sole | $m^3$ | $1.24 \times 10^{-5}$ | $5.36 \times 10^{4}$ | seJ/J | $2.79 \times 10^{6}$ |
| 77 | Water, unspecified natural origin | $m^3$ | $2.97 \times 10^{-5}$ | $3.06 \times 10^{4}$ | seJ/J | $3.80 \times 10^{6}$ |
| 78 | Water, well, in ground | $m^3$ | $6.34 \times 10^{-4}$ | $6.89 \times 10^{4}$ | seJ/J | $1.83 \times 10^{8}$ |
| | | | | **Emergy** | **seJ** | $\mathbf{5.94 \times 10^{12}}$ |
| | | | | **UEV** | **seJ/g** | $\mathbf{5.94 \times 10^{9}}$ |

\* Inputs quantities as reported in the ecoprofiles provided by Plastics Europe [40,65]. The nomenclature of the inputs was kept as shown in the original ecoprofiles to facilitate analysis and comparisons. \*\* UEVs used to calculate inflows' emergy were obtained from the literature and related to the baseline of $12.0 \times 10^{24}$ seJ/year [68,74].

Table 2 summarizes the results classifying the emergy inputs according to their contribution to the total emergy. Inputs were divided into crude oil, natural gas, energy, materials (mostly minerals and metals), and water. Regarding resource use, crude oil is clearly the main contribution to all products and, consequently, the main concern within the PET production chain. The contributions of natural gas go from 6% to xylene production to about 20% to PET resin. The energy supplied by different sources (including renewables, biomass, and hydroelectricity) contributes less than 6% to all products, and the contributions of water and materials are lower than 2% in all cases. Details of the relative contributions of emergy inputs are available in Table S9 in Supplementary Materials.

**Table 2.** Unit emergy values (in solar emergy joules per gram of product, seJ/g) of the PET production chain intermediaries were calculated and compared with literature. All values are relative to the baseline of $12.0 \times 10^{24}$ seJ/year [68,74], and mass/energy conversions used were $4.19 \times 10^{4}$ J/g for oil, and $5.13 \times 10^{4}$ J/g for natural gas [48,50].

| Product | UEV/($10^9$ seJ/g) This Work | UEV/($10^9$ seJ/g) Literature | Reference |
|---|---|---|---|
| Crude oil | 4.47 | 2.96 | [48] |
| Crude oil extracted | | 6.20 | [50] |
| Natural gas | | 2.80 | [48] |
| Natural gas extracted | | 6.66 | [49] |
| Naphtha | 4.52 | 5.02 | [49] |
| Xylene | 5.24 | | |
| Eethylene | 6.40 | 12.0 | [49] |
| p-Xylene | 6.15 | | |
| Ethylene oxide | 5.40 | | |
| Ethylene glycol | 3.99 | | |
| Purified terephthalic acid | 5.00 | | |
| PET resin | 5.94 | 26.1 | [60] |

Table 1. Summary of the LCI-based * emergy required to produce PET resin and nine selected intermediates (crude oil, naphtha, xylene, ethylene, p-xylene, ethylene oxide, ethylene glycol, and purified terephthalic acid). * The complete tables are available in the Supplementary Materials, Tables S1–S8.

Table 2 shows all the UEVs calculated in this study (see Supplementary Materials) compared with the UEVs from the published literature.

The LCI-based UEV of naphtha, for example, is close to the value estimated by Sha et al. [49], who used the value reported by Bastianoni et al. [48] for petroleum liquid fuels multiplied by the calorific value of naphtha. Similar results were obtained comparing the LCI-based UEVs with those estimated by [48,50] for crude oil. All UEVs vary within the same order of magnitude ($10^9$). Using the LCA database for all products may provide a more uniform inventory, which may partially minimize the problem of finding different values in the literature. The geographic location may have an influence due to the different kinds of electricity matrixes found in different countries. The variations in the calculated values may also vary according to the technological levels between the systems considered by the LCIs published by other authors. For example, Brazil's PET resin production process was an intensive consumer of biosphere resources with a much higher UEV value compared with that of European production [59,60].

### 3.2. The Potential Use of UEVs to Support CP Actions toward a Circular Economy

Higher UEVs values mean higher investment in environmental resources for the same amount of output. Still, UEVs may also be understood as measures of energy quality by expressing ratios between the quantities of the emergy required per joule of the product [45]. From this point of view, transformity provides a scale of energy transformations that establishes a hierarchy—the higher the transformity of a product, the more resources it requires and the more valuable this product is. The calculated UEVs for the PET production chain indicate that the efforts to introduce the chain into the circular economy focused on recycling postuse PET bottles are positive. They intend to keep the highest quality materials (highest UEVs) circulating as long as possible. Figure 2 shows an estimate considering the mechanical recycling beverage packages in which recycled PET reaches 55% of total PET, as foreseen by Plastics Recyclers Europe [40] for 2030. The emergy required for packing with recycled materials is significantly reduced compared to that which uses virgin materials at all stages of the production chain.

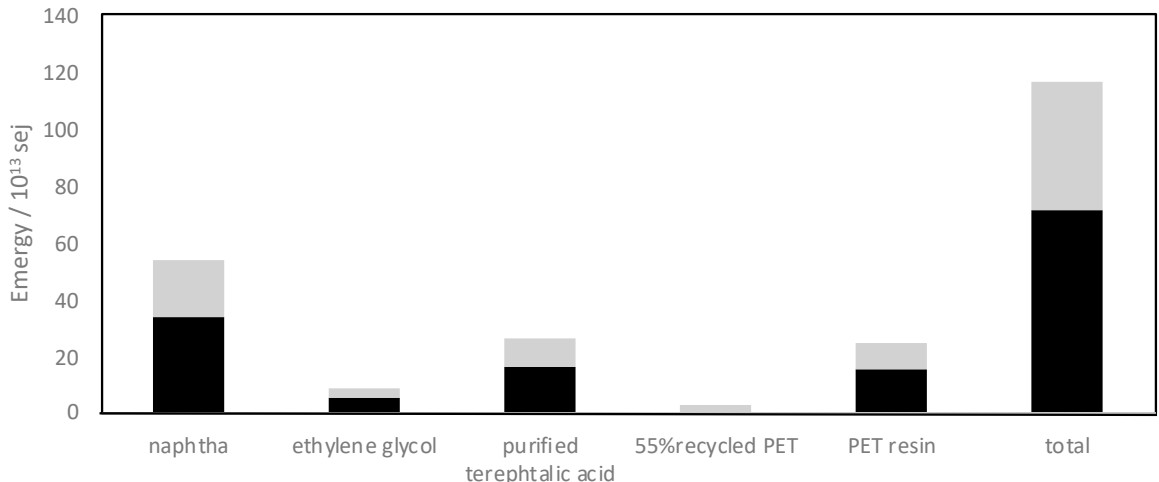

**Figure 2.** The potential effect of using 55 % of recycled PET on the emergy required to produce PET resin relative to 1000 L of beverage or 2 L PET bottles for the case of beverage packages mechanical recycling. Gray + black—without recycling, black—with recycling (for further details on different percentages of recycled PET, see [59,60]).

Despite the environmental load decrease in the overall life cycle, relative to the emergy use and the naphtha savings, it is crucial to prioritize/identify opportunities for improvements and changes considering the fast consumption of the final products in the case of PET packages [34], and the huge amounts of materials processed through the life cycle that continue to increase dissipative emissions. Shifting to a real circular economy for PET may help close the loop, preventing its way into landfills and protecting the marine environment against the (micro)plastic pollution [75].

## 4. Discussion

The obtained results highlight the importance of reliable databases for a complete and multidimensional sustainability assessment and more studies about the resource use to produce chemicals in the petrochemical production network. The discussion is divided into two main subjects: (Section 4.1) the importance of accurate and easy-to-calculate UEVs and (Section 4.2) how this indicator may help to improve the circularity in the PET production chain.

### 4.1. The Importance of Accurate and Easy to Calculate UEVs

UEV is a measure of environmental efficiency. Higher UEVs mean a lower environmental efficiency, which cannot be overlooked or ignored. This study sets up a foreseeable range for transformities of the petrochemical industry, reducing the uncertainties in selecting UEVs for the emergy analysis of the petrochemical industry and providing an alternative for future studies to expedite work on data collection. With the comprehensive data supplied by the Plastics Europe database [40,65], UEVs were calculated for PET resin and nine selected intermediates (crude oil, naphtha, xylene, ethylene, p-xylene, ethylene oxide, ethylene glycol, and purified terephthalic acid). These chemicals are also used in other industrial processes. Ethylene glycol, for example, is an industrial compound used as antifreeze, fluid for hydraulic brakes, solvents, paints, other plastics, and cosmetics.

Therefore, an easy UEV calculation routine based on elementary flows from the LCIs may provide consistent UEVs for the evaluation of several other processes and avoid allocation since every calculated UEV derives from the elementary flows that contribute to the product or intermediate from the extraction of natural resources to obtaining 1 kg of product "at gate" ready to be used in the following process.

The combination of LCA inventories and EmA helps quantify and qualify changes in resource use and facilitate policy adjustment, the adoption of a CP practice over another, and monitoring changes.

### 4.2. How UEVs May Help to Improve the Circularity in the PET Production Chain

The current most common supply chain was used to benchmark the system and evaluate CP options toward a CE.

Instinctively, the reuse option or the adoption of returnable bottles is generally accepted by the scientific community as better than recycling [76,77] since the more finished products circulate within their production chain, the closer this chain will be to the natural ecosystem, and the lower the load inflicted to the environment will be [78]. However, some authors advise against reusing PET bottles due to the potential release of phthalate esters with endocrine-disrupting effects [79].

Exploring a different angle, several routes for the synthesis of renewable para-xylene from biomass have been investigated with varying levels of accomplishment, proposing a renewable PET by obtaining both para-xylene and ethylene glycol from biomass [80,81], such as the Coca-Cola plant bottle, where ethylene glycol was partially obtained from sugarcane-derived bioethanol [82], or the paper bottle trial [83]. Both initiatives may reduce crude oil extraction by moving the PET production to another production chain based on agriculture, as shown in Figure 3. However, these two options should be deeply evaluated regarding their environmental performance without forgetting the debate on the use of soil with its priority for food production [84]. As a good start, Shah et al. [49] analyzed four

ethylene production systems—from naphtha, natural gas or pine residue, and switchgrass bioethanol—using emergy synthesis. The naphtha's UEV was 2–3 times lower than the UEV of the biomass-based ethylene, and naphtha steam cracking was distinguished by high yields and high loads on the environment. At the same time, processes based on biomass showed low emergy yield rates and lower environmental loadings. The trade-off lies between short-term profit and lower environmental loading ratios, but this kind of solution, severely accessed, may help decouple PET production from fossil fuel depletion.

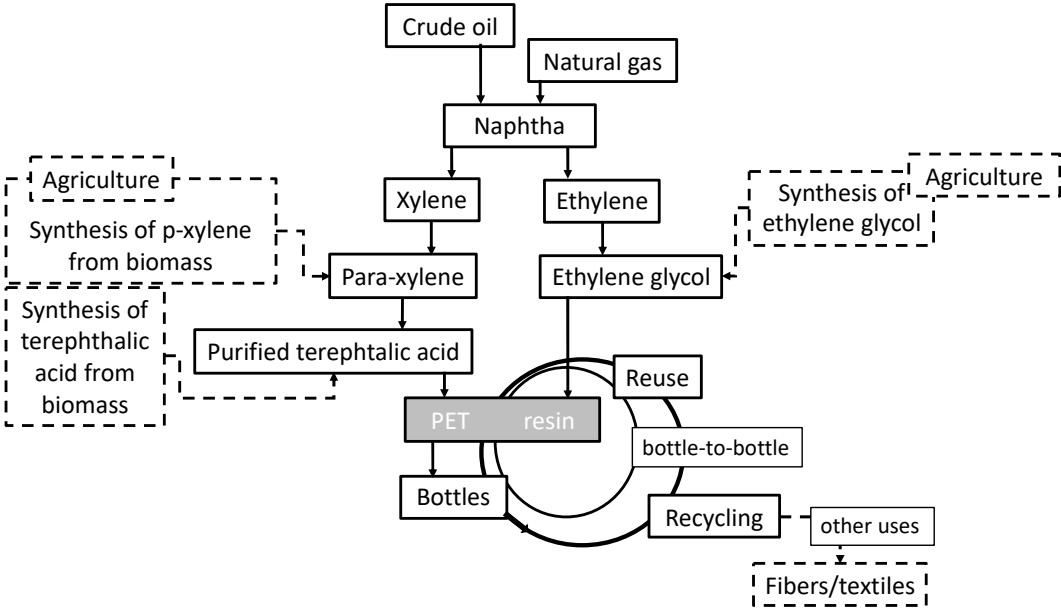

**Figure 3.** Potential routes to produce PET resin and some recycling options.

A third option lies in the chemical recycling [24,25] and PET decomposition into fibers or smaller molecules [27,28] to be used in different products such as the textile [85] or the construction industries [86]. These options extend the useful lives of these materials but also open new fields for research on the disposal of the new products, and despite recycling itself being an excellent option to promote a circular economy in the sector, much research and work still need to be done for its effective implementation through waste management strategies [87,88].

In this context, the UEVs results that show a clear gap between the transformity values of basic materials and those of their derivatives indicate that not only actions targeting these derivatives may be more effective to move from the linear "extract–make–dispose of" model to the circular one, but also selecting the best route comparing the environmental support demand and seizing the emergy savings due to the substitution of inputs, the combination of different supply chains, the reuse or recycling of waste or the effectiveness of management options. Figure 4 shows the different action fronts for implementing a well-planned CE that may expand the results from the PET resin production chain to other production systems.

Understanding the efficiencies (UEVs) in changing intra and inter practices, several alternative production chains help to assess the effect of cleaner production activities along each chain, the circulation of the materials at the interfirm or interchain levels, and the potential to increase the resilience of the overall system. This kind of assessment, taking into account all sectors depicted in Figure 4, may also support expanding existing synergies such as the reuse practices, the reduction of energy consumption, and the carbon footprint to transform the production chain from a linear to a closed loop. In addition, monitoring these efficiencies may also help guide research and the search for new technologies, establish and adjust public policies, manage alternatives, and promote awareness among consumers.

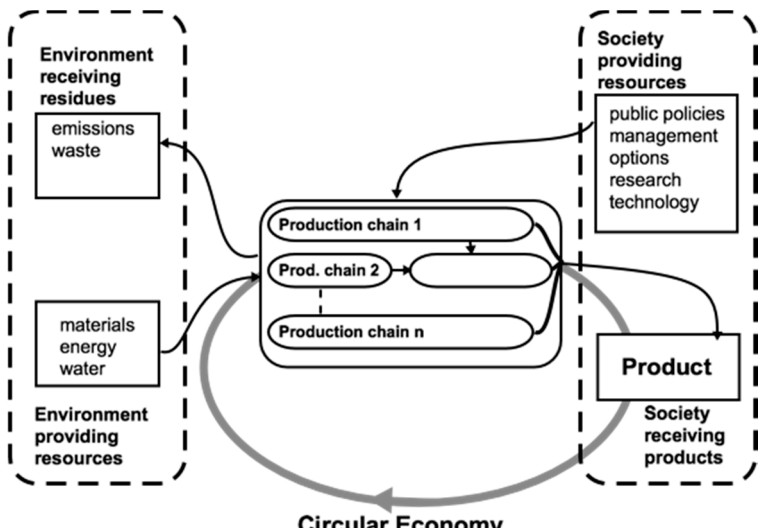

**Figure 4.** Required coordinated actions/assessments for the implementation of a CE. Inspired on the 5SEnSU model [89].

### 5. Limitations of This Study and Suggestions for Future Research

The limitations of this work relate to the intrinsic limitations of each method used—LCA and EmA—and there are still several research opportunities to develop theory concerning the combination of LCA and EmA. Despite its capability to account for the quantity of resources required by production systems, EmA is often criticized for its procedures, algebra, and reproducibility. In parallel, LCA is also criticized for depending on assumptions and scenarios through a simplified model that disregards the environmental impact of ecosystem services. Some researchers believe that the interpretation of the LCA system boundary lacks a scientific basis, which results in strong subjectivity in the system boundary, and there are errors or even contradictions in calculation results. Whereas emergy rules must be adapted to life-cycle structures, LCA should enlarge its inventory to give emergy a broader computational framework [90].

The combination of LCA and EmA allowed the inclusion of the value of resources provided "for free" by nature, expanding LCA comprehensiveness [91]. Introducing LCA databases to improve the accuracy of the analysis and compensate for limitations in EmA [91] may require the implementation of emergy algebra in the life cycle inventory and the expansion of the LCI system boundaries to include supporting systems usually considered by emergy but excluded in LCA (e.g., ecosystem services and human labor).

Future research should explore LCI databases, such as Ecoinvent, Open, LCA, and Plastics Europe, to provide UEVs that will help decision-making on circular initiatives in several sectors. The development of a combined tool considering the specificities of the production systems [50] to overcome both methods' limitations should also be addressed.

### 6. Conclusions

Although the benefits of an enforced circularity of the petrochemical sector are well-defined, multidimensional sustainability issues and the variety of options for the transition towards increased recycling and innovative materials call for indicators capable of assessing each proposal and prioritizing actions toward closing loops, discerning the quality of energy used and the processes' efficiency.

The PET production chain was explored from the standpoint of resources and energy use, covering a selection of intermediate products. New UEVs for petrochemical commodities were estimated using LCIs databases (xylene, para-xylene, ethylene oxide, ethylene glycol, and purified terephthalic acid), establishing priorities and providing an indicator to interfere in the PET production chain.

Moving to a real circular economy is complex and cannot rely on single life cycles of chemical products but instead on the chemistry sector as a whole, combining different production chains to replace the current linear approaches. For this, operational barriers to co-work among different sectors, such as the biobased plastics sector and the petrochemical one, both supported by public policies and consumer awareness, encourage the transition to a circular, sustainable, and well-designed production network. There is a need for this to be focused on given the short useful life of most plastics—packages in particular—and their latent damaging impacts on the environment. The shift towards a circular economy for the plastics sector in which wastes are either reused, remanufactured, or recycled into old/new uses, compels coordinated actions. One of them is the understanding of the efficiencies (UEVs) changing across all life stages and all life cycles so that shareholders can make well-versed judgments and choices as to where and when CP interventions are most effective.

The complementarities of the two methods were helpful in providing a complete comprehension of the system organization, and the joint use of the methods offered important elements and valuable information for the understanding of the organization of the PET productive system and the use of the materials and energy flows that determine its development and operation.

**Supplementary Materials:** The following supporting information can be downloaded at: https://www.mdpi.com/article/10.3390/su14116821/s1, Table S1. LCI-based emergy required to produce 1 kg of crude oil. Table S2. LCI-based emergy required to produce 1 kg of natural gas. Table S3. LCI-based emergy required to produce 1 kg of naphta. Table S4. LCI-based emergy required to produce 1 kg of pygas. Table S5. LCI-based emergy required to produce 1 kg of xylene. Table S6. LCI-based emergy required to produce 1 kg of ethylene. Table S7. LCI-based emergy required to produce 1 kg of p-xylene. Table S8. LCI-based emergy required to produce 1 kg of ethylene oxide. Table S9. LCI-based emergy required to produce 1 kg of ethylene glycol. Table S10. LCI-based emergy required to produce 1 kg of purified terephthalic acid.

**Author Contributions:** Conceptualization, C.M.V.B.A.; methodology, G.B.; formal analysis, F.A.; writing—original draft preparation, G.B.; writing—review and editing, B.F.G. and G.L. All authors have read and agreed to the published version of the manuscript.

**Funding:** This research received no external funding.

**Acknowledgments:** The authors acknowledge the support of Vice-Reitoria de Pós-Graduação e Pesquisa da Universidade Paulista (UNIP).

**Conflicts of Interest:** The authors declare no conflict of interest.

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
