# Peer review of "Prioritizing Cleaner Production Actions towards Circularity: Combining LCA and Emergy in the PET Production Chain"

_sustainability, doi:10.3390/su14116821_

Round 1
Reviewer 1 Report
I have made the suggestions and comments directly in the manuscript attached herewith. Revise the paper accordingly.

Author Response
The authors sincerely thank the reviewer’s comments and suggestions that helped improve the text and clarify the ideas to be exposed.
We inform you that all reviewers' comments and suggestions were considered. Please see our answers to each one in blue.
Reviewer 2 Report
Dear authors,
As one of the selected reviewers I evaluated your submission. I have no special issue with your analysis and method, but I found some serious structural issues that must be fixed before proceeding. firstly acronyms do not used in teh title and must be acknowlwdged in full words when refer in the title.
Then your abstract is too general. you must rewrite it so that it includes research problem, method, main findings and main contribution.
I couldn't understand your main problem statement too.
your conclusion is also too general. I suggest you to express your contribution comapring with previous researches.
Also I recommend you to add twio sections of "suggestions for future researchers" to recommend how they can contunue your work and "research limitations" to show level of generalization of your results.
I look forward to receive the revision of your submission.
Best of luck

Author Response

(The authors gave the same response as above.)

Reviewer 3 Report
- Line 80, there is an extra bracket symbol.
- Line 104-107, the font size is inconsistent.
- Reference number should be added in Table 3.
- The novelty should be highlighted. Most figures are from literature. It would be good to provide more quantitative analysis based on the calculation.
Author Response

(The authors gave the same response as above.)
